# Managing risk in cancer presentation, detection and referral: a qualitative study of primary care staff views

Neil Cook,[1] Gillian Thomson,[2] Paola Dey[1]

[1]School of Medicine and Dentistry, University of Central Lancashire, Preston, UK
[2]School of Health, University of Central Lancashire, Preston, UK

**Correspondence to**
Neil Cook;
ncook2@uclan.ac.uk

## ABSTRACT

**Objectives:** In the UK, there have been a number of national initiatives to promote earlier detection and prompt referral of patients presenting to primary care with signs and symptoms of cancer. The aim of the study was to explore the experiences of a range of primary care staff in promoting earlier presentation, detection and referral of patients with symptoms suggestive of cancer.

**Setting:** Six primary care practices in northwest England. Participants: 39 primary care staff from a variety of disciplines took part in five group and four individual interviews.

**Results:** The global theme to emerge from the interviews was 'managing risk', which had three underpinning organising themes: 'complexity', relating to uncertainty of cancer diagnoses, service fragmentation and plethora of guidelines; 'continuity', relating to relationships between practice staff and their patients and between primary and secondary care; 'conflict' relating to policy drivers and staff role boundaries. A key concern of staff was that policymakers and those implementing cancer initiatives did not fully understand how risk was managed within primary care.

**Conclusions:** Primary care staff expressed a range of views and opinions on the benefits of cancer initiatives. National initiatives did not appear to wholly resolve issues in managing risk for all practitioners. Staff were concerned about the number of guidelines and priorities they were expected to implement. These issues need to be considered by policymakers when developing and implementing new initiatives.

## Strengths and limitations of this study

- The study investigates the experiences of a range of primary care staff around implementing initiatives for the earlier diagnosis of cancer.
- The underlying concern in primary care is related to managing risk.
- The sample included a mix of practices with different practice characteristics and included those known to be engaged in national and regional initiatives and those who were less engaged.
- The sample was drawn from a single English region.

Nationally and internationally, primary care is seen to have a key role in improving cancer survival by reducing delays in diagnosis through promoting earlier presentation and through earlier detection and referral of those with symptoms for further specialist assessment.[3 4] In the UK, national campaigns exhort those with symptoms to see their general practitioner (GP) earlier; there are national referral guidelines for suspected cancer and a national system for urgent referral from primary to secondary care (2-week waiting-time initiative).[5] While these appear to have contributed to improved survival, evidence also suggests that there may be further room for improvement[6]: there is practice variation in the use of the '2-week' initiative and some patients are seen several times in primary care before referral.[7–9] A significant proportion of patients also present through emergency routes and have poorer survival.[10]

To further support primary care in the UK, resources were developed and/or disseminated by the Department of Health's National Awareness and Early Diagnosis Initiative (NAEDI), established in 2008, and the Royal College of General Practitioners (RCGP) including: audit and significant events analysis tools,[7 8] GP level cancer profile data, safety netting recommendations and risk assessment tools.[11] Key initiatives have mainly focused on GPs, but other members

## INTRODUCTION

Studies in the 1990s showed that UK cancer survival rates were worse than many other European countries, following which there has been two decades of concerted effort to expedite access to proven effective cancer treatments.[1] While there have been improvements in cancer survival latterly, the UK still lags behind other countries with similar healthcare systems, which may be partly due to later stage of disease at presentation.[2]

of the primary care team may have key roles: an area which has been largely unexplored.

A number of international studies have been undertaken in attempts to understand the reasons behind delay in cancer diagnosis. Qualitative studies have mostly focused on patient perspectives.[12–14] Only a few have explored primary care experiences and these have been mainly limited to decision-making processes or referral pathways[11 15 16] and from the perspective of the GP.[3 11 15 16] In order to understand how a range of initiatives across the patient pathway in primary care could be more effective and the role of other members of the practice team, we explored the experiences of a range of primary care staff in supporting earlier presentation, detection and referral of those with symptoms suggestive of cancer. The study was undertaken in one region of England, which at that time was covered by the Lancashire and South Cumbria Cancer Network (LSCCN).

## METHODS

This was a qualitative descriptive study utilising individual and group-based interviews. It aimed to recruit staff from six practices who were differentially engaged with the national awareness and early diagnosis of cancer agenda. GP practices within LSCCN were stratified into one of three groups at the end of June 2012. High engagers had participated in at least one of the following: RCGP cancer audit, attendance at a course on early diagnosis or face-to-face meetings to discuss GP cancer profiles and action planning. Medium engagers had attended at least one regional meeting about cancer awareness. Low engagers were not known to have engaged in any initiatives. Of 254 practices in the geographical area, there were 51 high engagers, 69 medium engagers and 134 low engagers.

Within each group, a random sample of 10 practices were sent a letter about the study, followed up with a phone call from the research team 1 week later. The initial aim was to recruit at least two practices in each stratum. Owing to low uptake within the medium and low engager strata, these categories were merged and a random sample of a further 15 practices were sent letters. Practices were offered a choice of either group or individual interviews as it was recognised that time constraints prevent some practice staff from taking part in group interviews and some may feel uncomfortable discussing the issues with colleagues. Five practices agreed to a group interview, with one group interview undertaken in each practice. Individual interviews were the preferred method in only one practice, with four interviews undertaken in this setting.

The semistructured, audiorecorded interviews, which were attended by two researchers, occurred between September and October 2012. We recruited three practices in the high stratum and three practices in the merged medium and low strata (a mix of low and medium engagers). The topic guide is outlined in box 1. Thirty-nine participants took part in the study; 35 took part in one of the five group interviews and four took part in individual interviews. Job roles included GP (n=9), receptionist (n=7), nurse (n=6), manager (n=6), secretary (n=5), healthcare assistant (n=3), medical student (n=2) and phlebotomist (n=1). In each practice, GPs, other clinical and administrative staff were involved in the interviews. Practice characteristics are shown in table 1. Further detail at practice level is not presented due to the level of confidentiality agreed with our participants.

The interviews took between 38 and 67 min to complete, and between 4 and 11 staff took part in each practice. All transcribed data were entered into NVivo V.10 and analysed using a thematic network analysis approach.[17] This involved an iterative and cyclical process of reading and analysis to identify basic, organising and global themes within the dataset. Analysis was undertaken by two authors (GT and NC) independently on six transcripts initially, followed by an in-depth discussion and consensual validation of key themes. A further cycle of independent and collaborative analysis was then

---

**Box 1** Semistructured interview schedule

▶ Why do you think people with symptoms suggestive of cancer do not present, get seen or diagnosed or referred earlier?
▶ What do you know about the initiatives that concern the earlier presentation, diagnosis and referral of cancer symptoms?
  – Prompts around specific tools/initiatives/referral criteria
▶ Have you accessed any local training events concerning the identification, referral of patients with cancer?
  – Prompts around specific training attended, and how learning is usually undertaken
▶ Overall, what is working well in terms of the implementation and use of these initiatives?
▶ Overall, have you experienced any/or what do you consider to be the main barriers in the implementation/access/use of these various initiatives?
▶ Are there any practice-based issues that may affect the early identification and referral for patients with cancer?
  – Prompts around staff and practice issues, communication, administration
▶ Overall, what do you think local practices could do to help promote and diagnose cancer symptoms?
  – Prompts around team roles and other practice issues
▶ Do you know about the Lancashire and South Cumbria Cancer Network and what their role is?
  – Prompts around extent of engagement, attitudes and areas for improvement
▶ What is your opinion concerning the forthcoming 'access to diagnostics' initiative (initiative for practices to make direct referrals to diagnostics such as X-rays, CT scans, ultrasound)?
  – Prompts around who should make the referral
▶ Are there any additional support mechanisms/external to the practice that need to be in place to help with promotion/diagnosis and referral?
▶ Any further issues or concerns you would like to raise about this work?

**Table 1** Practice and interview characteristics

| Practice characteristic | Number of practices |
|---|---|
| Number of partners | |
| 4 or more | 3 |
| 3 or less | 3 |
| Deprivation quintile* | |
| 5 or 4 (more deprived) | 3 |
| 2 or 3 | 3 |
| 1 (most affluent) | 0 |
| Population density† (%) | |
| <15 | 2 |
| 15–39 | 2 |
| ≥40 | 2 |

*Based on index of multiple deprivation of practice location 2010 (source: Department for Communities and Local Government, Indices of Deprivation 2010).
†Person per hectare based on practice location (source: Office for National Statistics (ONS) Neighbourhood Statistics).

undertaken on a further subset of transcripts to ensure rigour and authenticity of the themes generated. All thematic decisions were discussed with the third author (PD). Individual written informed consent was obtained from all participants.

## RESULTS

Overall, practice staff were well aware of the 2-week waiting-time initiative and had good knowledge about the national 'Be Clear on Cancer' cancer awareness public campaigns, but had less awareness about other initiatives specifically targeted at primary care. The key global theme to emerge from the interviews related to 'managing risk' within primary care:

> It's quite a tricky, nebulous area […]. The nature of general practice is that we're dealing every day with uncertainty… (Interview 6, Participant 3)

A key concern of staff was that policymakers and those implementing cancer initiatives did not fully understand how risk was managed within primary care. Cancer was only one priority and there was an abundance of initiatives for a variety of conditions which primary care staff were expected to implement. Three underpinning organising themes (and associated basic themes) of 'complexity', 'continuity' and 'conflict' highlighted the tensions and difficulties that primary care face in managing the risk of early detection and referral for cancer symptoms while dealing with complex symptoms and care systems, patient-led factors and target-focused care. An overview of the organising and basic themes is presented in figure 1. These themes are described and discussed, contextualised by participant quotes below.

## Complexity

This theme highlighted the complexity of managing risk in early cancer diagnosis because of external factors including cancer symptom differentiation and the restrictions imposed by referral criteria; the multitude of services and professionals involved in diagnostic and assessment services and the plethora of policies and initiatives targeted at primary care practice.

*Cancer disease and symptoms*: Cancer diagnosis was an important priority area in primary care but diagnosis presented complex challenges. Cancer was a rare diagnosis in primary care although symptoms associated with cancer were common.

> We get lots of sore throats, and yet we get one tonsillar cancer every three [years] so sorting out the wheat from the chaff is a real challenge. (Interview 3, Participant 3)

These complexities of diagnosis were compounded by what were considered rigid referral criteria, based on disease prevalence among those with symptoms, which led to cases being 'bounced back' if they failed to meet diagnostic criteria. In attempts to manage what they considered was a risk to patients, some participants said, on occasion, they had to 'fudge', 'embellish' or 'bend-the-rules' to ensure assessment of patients for whom they had concerns:

> Even though it says on the form, "don't fill out this form unless they tick any boxes" but you find a box to tick, and usually for very good reason. And I think you'd only go slightly over egging the presentation if you were pretty sure there was something there, something going on. (Interview 4, Participant 4)

*Fragmentation and access to diagnostic services*: Primary care staff expressed difficulties accessing diagnostic services due to ongoing service reconfigurations and the involvement of multiple agencies. This led to fragmentation in terms of staff not always knowing who, or to which services, referrals could be made.

There were also some concerns about fragmented relationships between primary and secondary care with several participants feeling frustrated by restricted access to diagnostics for certain conditions. Some participants considered this was due to 'empire building' by professionals justifying and 'preserving' their service by retaining ownership of who was qualified to make referrals:

> I can send people for a CT scan if I have a concern about them, for some conditions, but not others. Well why not? I'm the one who's initiating the referral in the first place. You trust me to initiate the referral to pick the patient and prepare them so that you can come along and just arrange the scan and look at it. (Interview 4, Participant 3)

While some GPs wanted more direct access to diagnostics, this was not universal, and others highlighted the need for 'training' prior to referrals being made.

*Guidelines content and information overload*: Participants referred to the usefulness of guidelines to help symptom differentiation and manage risk. Knowledge was felt to

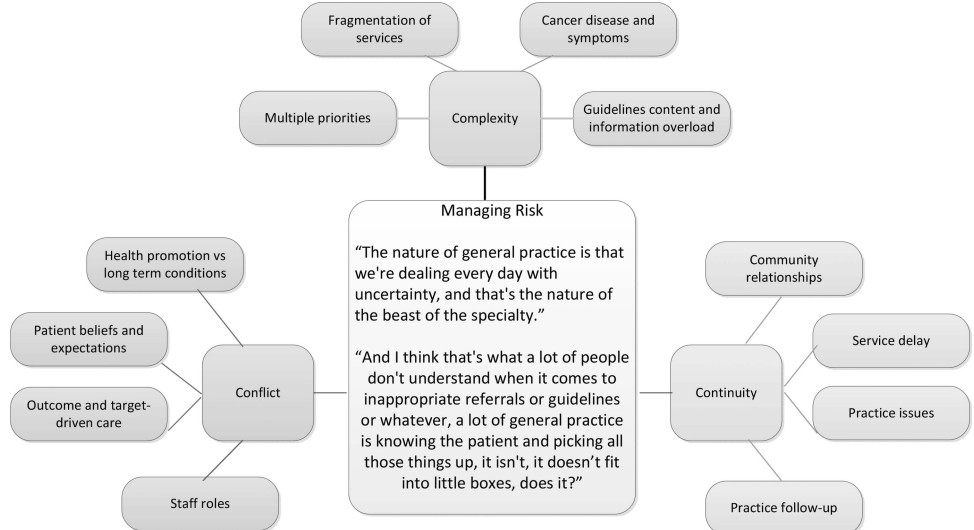

**Figure 1** Organising themes of complexity, continuity and conflict with their associated basic themes, centred around the global theme of managing risk in primary care.

be experientially derived and referral guidelines only considered useful for less experienced doctors. Risk assessment tools were occasionally considered unnecessarily complex when patients had 'red flag' symptoms and, as previously mentioned, sometimes restrictive for use in primary care:

> I mean, you see, from a GPs point of view, if somebody comes in who is fifty-five years of age, and passes blood in his stool, you need to exclude a cancer. Now, I don't want to know how many percentage of those people pass blood in the motion will have cancer, or how many will have piles, or inflammatory bowel disease or what have you. (Interview 4, Participant 3)

Concerns were expressed about the number of guidelines available for different conditions and inconsistencies between different sources:

> And that's the problem 'cause last year there was a big campaign, "if you have a cough for more than three weeks you need to go and see a doctor", but NICE guidelines is six weeks. (Interview 6, Participant 4)

Participants described how the 'tsunami' of new guidelines, care plans and initiatives meant it was difficult to keep up-to-date:

> And I thought 'that's probably a squamous cell carcinoma. That needs a two-week wait referral'. Then, I actually checked the guidelines, the guidelines say that "it's got to be over one centimetre and you've got to wait more than eight weeks really for it to grow" etcetera, etcetera [...] and those things will change on a regular basis. Now, how do you expect an entire network of primary care physicians to stay current with all those guidelines and apply them religiously? (Interview 2, Participant 3)

## Continuity

Practice staff expressed concern that a lack of continuity increased the risk of missing diagnoses and/or supporting the patient through a difficult period. This was highlighted through discussions about relationships between patients and clinicians, delays in information sharing across the primary and secondary care interface and patient follow-up after initial consultation.

*Patient–practice relationships*: Continuity of carer could not always be achieved in practice, although it was felt that patients preferred it and its absence was thought to possibly contribute to diagnostic delay:

> People have been hopping around from one clinician to another and that, and you don't see the evolution of the story until it hits you in the face. Patients book on the day, they don't necessarily get in the person they normally see, they get in with the available, so that can disjoint things. (Interview 3, Participant 4)

Staff felt that smaller, rural practices had closer relationships with their patients and their community, which would lead to earlier consultation. However, others felt awareness of cancer deaths was heightened in close-knit communities, which could reinforce negative views towards cancer and cancer survivorship among the local population and delay access to services.

*Primary and secondary care interface*: Expedited assessment following GP referral of patients through the 2-week waiting-time initiative was perceived to work well by practice staff, but delays further on in the assessment process, and outside of primary care control, were felt to put patients at risk again and exacerbated their concerns:

> And, well, it's very difficult from our point of view, 'cause we're saying, "look, you know this may be nothing serious, however, I want to refer you on the system, you'll

get an appointment in two weeks", and they do get seen, but then there's a massive delay, so, you know, then they're kinda all heightened up because they're thinking, "{whispered} the doctor thinks I've got cancer" and then nothing happens for ages, and it's really hard for the patients. (Interview 3, Participant 5)

*Practice-based follow-up*: Patients themselves were also considered to cause delays if they failed to attend. Practices debated the extent to which they should chase these patients up and the risks if they did not:

In the back of their mind they know they should, but if they face the fact that they're going back, are they are going to be faced with something else? You also have to be responsible for the fact that if that doesn't stop, you must persist, and accept the fact that, if you don't, then you could, somewhere down the line end up with something so serious that it's only going to be palliative. (Interview 2, Participant 1)

## Conflict

Within this theme, managing risk was related to primary care focus on targets, conflicting opinions about the role of non-clinical staff and about the worries and tensions generated by cancer awareness campaigns.

*Prevention* versus *target-driven care*. Primary care staff considered cancer a public health priority but some felt that there was a tension between adopting preventative approaches in the practice such as promoting cancer awareness and early presentation and the way that primary care is currently funded through achieving specific targets mainly relating to the management of long-term conditions (Quality and Outcomes Framework):

We're not as good with public health, with local public health initiatives, as maybe we should be necessarily. It's just time to do things, and those things aren't required of our core business, and when you have a very detailed contract that tells you what you will be paid for doing. (Interview 2, Participant 3)

*Staff roles*: There were conflicting views around the role of other practice staff in managing the risk of cancer detection and awareness, in particular with regard to reception staff. Some staff viewed the reception role as a health advisor, whereas others perceived their role to be purely administrative. Additionally, there were reservations about the ability of reception staff to relay cancer messages and how the public might feel about receiving health information from them:

You're passing clinical responsibility to people who are non-clinical. It's alright for patients to speak to them, and that's fine, but, I'm not gonna use the receptionists as a source of information […]. I think anything clinical should be passed to the doctor full stop. I don't think there is any other role for the receptionist. (Interview 4, Participant 3)

There were also concerns about potential litigation issues if reception staff were to give advice and how this could impact on the practice:

I think it's also worth mentioning at this point, because of our roles, how they are, we get a lot of patients not happy about us supplying information, because we're not allowed to diagnose, obviously for obvious reasons, but if you was to imply something, that could come back on us twice as hard because you'd implied something that could be wrong, and therefore it is now your fault. (Interview 1, Participant 1)

A further issue associated with staff roles in helping to manage risk was the identified benefits of having a chain of communication between all practice members. For example, receptionists being able to 'raise issues' with the doctor if they know someone is coming in who has symptoms the doctor should be aware of but the patient 'may not say anything'. This can also work the other way:

Patients that I worry about, I will leave messages. I'll mention to reception or mention to people that I want to follow them up. (Interview 6, Participant 3)

Others felt that the focus on the long-term conditions might provide opportunities for practice nurses to be more involved as they were often seeing patients who might be at higher risk of cancer because of their age, disease condition or lifestyle behaviours:

Chronic disease, yeah, so diabetic, high blood pressure, chronic kidney disease, asthma, most patients will come and see the practice nurses for routine bloods, blood pressure, weight, everything like that. So often, GPs don't always see them, I mean they do the medication reviews, but, we flag to them anything that we're worried about. (Interview 2, Participant 4)

Some could also see a role for members of the wider practice team:

We'd say 'well who are the district nurses already going out to?' You know, you can work with your local pharmacists. (Interview 2, Participant 2)

*Cancer awareness*: Although cancer fear was commonly acknowledged, staff were divided as to whether patients were afraid to consult the GP with cancer symptoms. Some felt that unhealthy lifestyle choices contributed to patient reluctance to present with cancer symptoms:

The cancers where they feel that they may have contributed to it, like smoking, they tend to ignore because they don't want to be told that it may be their fault in a certain way, and they don't want to give up the lifestyle. (Interview 6, Participant 4)

Cancer awareness campaigns were felt to be important, although those who presented to the practice

following the campaign were more likely to be those at least risk:

> It's the people who never come to the doctors that you want to hit, not the people that were already coming in anyway, and they tend to be ones that see those adverts. But then if you can get one person who wouldn't normally come in and you catch them, then it's better. (Interview 5, Participant 1)

Campaigns were also felt to create extra work for all practice members. There were complaints that cancer awareness campaigns tend to run without consideration of other local or national campaigns which may be running simultaneously which increased the risk that practice capacity to safely respond to patients was compromised:

> It's like it's alright saying putting in place to get them in, but it's the sheer volume, isn't it? You can only cope with so much can't you? (Interview 5, Participant 1)

## DISCUSSION

This study provides insights into the experiences of primary care staff who manage patients with symptoms associated with cancer. The overarching theme that emerged was the need to manage risk so that patients with cancer had a timely diagnosis and were assessed appropriately. The associated subthemes of complexity, continuity and conflict highlighted the tensions and difficulties faced by staff when attempting to manage these risks in modern practice.

The study was small, descriptive and exploratory. However, it covered a diverse geographical area and included practices with varying levels of engagement with awareness and early diagnosis initiatives. We undertook stratified sampling of the practices to ensure we had a balance of perspectives so as to better inform policymakers of the possible barriers and drivers to the uptake of initiatives. The stratification was based on known engagement with a range of national and regional initiatives to promote earlier presentation, detection and referral of patients with symptoms suggestive of cancer available to the practices at that time.

The findings of our study suggest that national initiatives did not appear to wholly resolve issues around managing risk for all practitioners. Rather, in some cases, these initiatives were felt to introduce inherent risks which staff had to find ways to overcome. For example, national cancer awareness campaigns were felt to be very important to encourage patients who would otherwise delay diagnosis. However, as these campaigns appeared to increase consultation rates from those at lesser risk, this was perceived to place additional burden on the practice.

There were also concerns among our participants about practice-based initiatives. The International Cancer Benchmarking Partnership observed lower survival rates in Denmark and the UK compared with other countries

in their study, and raised concerns about the role of primary care gatekeeping in delayed diagnosis.[2] This observation appears to be supported by a wider study of European countries which also demonstrated lower cancer survival in those with primary care-based gatekeeper systems.[18] Primary care gatekeeping may reduce the burden on specialist care but may also contribute to diagnostic delay through the restriction of access to these services. In the UK, national referral guidelines and the 2-week waiting-time initiative should help alleviate this problem for those with symptoms associated with cancer. Participants in our study considered that timely diagnosis was as an essential part of their role in cancer care. An Australian study had a similar finding,[16] but highlighted that the resources spent gaining timely access to specialist opinion were a major issue for their practitioners. In our study, this was universally considered less of a problem because of the 2-week waiting-time initiative, but concerns were expressed that lack of direct access to diagnostic investigations and poor communication between primary and secondary care put patients at risk of extended delays. Similar concerns were reported in a study undertaken in Ireland.[3] Some participants in our study felt that the referral criteria for the 2-week waiting-time initiative were too restrictive. This led to practitioners, on occasion, subverting the referral system to ensure that patients they considered were at risk, but who did not fit the referral criteria, could be assessed in a timely manner. Such concerns are not unfounded; one study has shown that 8% of patients felt by GPs to have cancer, but who did not have symptoms which fit referral criteria for cancer, were subsequently diagnosed with cancer.[19] This may be because the presenting signs and symptoms had a lower predictive value for cancer than those included in the guidelines. Nevertheless, there is high compliance with guidelines and some limited evidence that referral guidelines contribute, in part, to diagnostic delay reduction.[6 19]

Diagnostic errors leading to primary care malpractice claims are common.[20] Researchers suggest that although GPs are more likely to correctly diagnose patients with cancer than miss cases, there are a disproportionate number of deaths among the latter.[21 22] Diagnostic complexity is compounded by the frequency of consultations for symptoms associated with cancer,[22 23] a fear highlighted by practitioners in this study. A study in Norway, which followed up patients presenting with cancer warning signs to their GP, suggests that patients with cancer may be missed if multiple warning signs and symptoms are not considered.[22] Retrospective studies of practice-based data have identified combinations of warning signs and symptoms which may be associated with increasing likelihood of cancer.[24 25] These have been used to inform risk assessment tools to aid decision-making, with algorithms and probabilities of risk based on demographic characteristics, lifestyle factors, symptoms and/or attendance frequency.[11] Some have been disseminated nationally, and there is evidence from

quantitative and qualitative studies that they affect GPs' decisions to refer.[11] However, in our study, some participants had similar concerns about these tools as they did about guidelines. A recent analysis of significant event audits in lung cancer highlighted how cancer can mimic other diagnoses and have atypical presentation.[7] Guidelines and risk assessment tools are analytical tools, and although statistical probabilities are based on uncertainty, some participants in our study felt they may introduce a level of certainty which fails to encapsulate practitioners' tacit concerns about patients. Previous work in primary care decision-making has emphasised the analytical over the experiential, but recent research, including in cancer diagnosis, suggests that experiential knowledge may have a role and may be more responsive to the patient as a person.[15 26–28] Others have found that better diagnostic decisions about urgent referrals appear to be made by older doctors[23] and, as commented on by some of our participants, that guidelines and risk assessment tools may be more useful for newer practitioners who have yet to develop problem-solving strategies.[29]

It has also been observed that, unlike in acute care, decision-making in primary care partly involves an understanding of the patient context and perspective.[30] Our study participants highlighted concerns that lack of continuity of carer may lead to diagnostic delays. Relational continuity was felt to be a particular problem in larger practices and urban settings. Other studies suggest that fragmentation of primary care and shorter consultation times has led to a lack of continuity of care or carer, which may hinder early diagnosis as cancer presentation can be complex in those with comorbidities.[16 31 32] In a qualitative study of patients with lung cancer in New Zealand, patients felt that not always seeing the same GP could lead to delays in diagnosis due to poor follow-up.[33] In a study in Denmark, the authors suggest that perceived lack of accessibility and doctor–patient relationship were associated with patient delay in seeking advice about cancer symptoms.[34] Others have found that confidence and trust in a doctor were more important predictors of cancer detection than ease of access and choice of preferred doctor.[35]

In our study, participants highlighted how other practice team members, such as nurses who are in more regular contact with patients with long-term conditions, could raise symptom awareness or be more alert to symptoms suggestive of cancer. In the USA and Canada, patient navigators are being used to support patients through the complex systems found in cancer management and care.[36] It has also been suggested that these roles could be extended to support patients during the diagnostic, referral and assessment processes and ensure appropriate follow-up of investigations.[37] In our study, some staff felt that receptionists could act as navigators in terms of directing patients with symptoms to see the GP. Low levels of knowledge about some cancer symptoms have been demonstrated in non-medical staff.[38] However, our study highlighted that professional

boundaries and concerns about litigation might impede such initiatives. Disclosures or discussions of diagnostic issues within a public reception location also raised ethical concerns. Research within primary care suggests that facilitation of communities of practice and interdisciplinary knowledge sharing may help to identify the role of other practice team members in promoting earlier cancer presentation and diagnosis,[39] although training needs should be addressed.[38]

This study has highlighted that a 'one-size-fits-all' approach to implementing initiatives is unlikely to succeed as practitioners expressed a range of views and opinions on the benefits of different initiatives. Lack of implementation of initiatives in primary care is not always because of resistance to the initiatives themselves, but sometimes because of the sheer number across a range of priority areas that practices are expected to implement, often simultaneously. Policymakers should consider more carefully how these impact on primary care, how they can be embedded into practice systems and emphasise and exploit synergies with other disease conditions.

For there to be greater success of initiatives aimed at promoting earlier presentation, detection and referral in primary care, there needs to be further work on understanding how primary care manage risk in the face of inherent uncertainty, organisational changes and competing priorities.

**Acknowledgements** The authors would like to thank the practices who participated in this study and to the staff of the Lancashire and South Cumbria Cancer Network who assisted them.

**Contributors** NC recruited participants, carried out interviews, transcribed the interviews, coded the transcripts, contributed to the development of the coding framework and contributed to the writing of the manuscript. GT carried out interviews, coded the transcripts, contributed to the development of the coding framework and contributed to the writing of the manuscript. PD designed the study, collated practice engagement data, contributed to the development of the coding of the framework and contributed to the writing of the manuscript.

**Funding** This work was supported by Lancashire and South Cumbria Cancer Network (LSCCN) as part of the National Cancer Awareness and Early Detection Initiative (NAEDI) funding 'cancer networks supporting primary care'.

**Competing interests** PD was public health lead for LSCCN and NC is part-funded by a grant from LSCCN. PD was also funded by LSCCN to undertake evaluative work.

**Ethics approval** University of Central Lancashire: Science, Technology, Engineering, Medicine and Health (STEMH) ethics committee.

**Provenance and peer review** Not commissioned; externally peer reviewed.

**Data sharing statement** No additional data are available.

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
