## [Reviewer comments · BMJ Open]

Some articles will have been accepted based in part or entirely on reviews undertaken for other BMJ Group journals. These will be reproduced where possible.

ARTICLE DETAILS

TITLE (PROVISIONAL)	Managing risk in cancer presentation, detection and referral: a qualitative study of primary care staff views
AUTHORS	Cook, Neil; Thomson, Gillian; Dey, Maria

VERSION 1 - REVIEW

REVIEWER	May-Lill Johansen Assistant Professor Department of Community Medicine, UiT The Arctic University of Norway
REVIEW RETURNED	02-Mar-2014

GENERAL COMMENTS	1. In the abstract, the objective is stated as: To explore the barriers and facilitators, in primary care, to early detection and prompt referral of patients with symptoms suggestive of cancer. However, the introduction, p 4 line 30, says that "we need to understand the barriers and facilitators to the promotion of earlier consultation, detection and referral" and in line 40:"we focus on the experiences of implementing cancer detection and referral initiatives and the perceived barriers and facilitators to earlier consultation, detection and referral of those with symptoms suggestive of cancer." Thus it is unclear whether the focus is on the experience of the initiatives, the experience of implementing the initiatives, or the experience of diagnosing and referring for cancer i primary care. 'Barriers' and 'facilitators' are in any case analytical categories, which do not become clear in the present analysis nor in the discussion, especially not 'facilitators'. 2. The last sentence in the result section of the abstract,p 2 line 50-54, is not clear enough to the first time reader. What is your message? The conclusion of the abstract, p 3 line 5, draws on aspects that was not mentioned in the abstract result section. 8. See the comments at the bottom. 9. Barriers and facilitators are not mentioned as analytical categories in the results section nor in the discussion. This could be solved by changes the objective to "explore experiences..." 10. The key global theme "managing risk" is a strong analytical category. However, it is not explained and elaborated well on. Here, the authors should build a strong line of arguments linking the analytic network together. At present, the rest of the result section is weak. There are too many
--

long citations and too little analytical work presented. The citation on page 6 line 35 does not relate to the comments above. The authors' comments between the citations contain several in vivo concepts that should be elaborated on to create a deeper understanding of what the respondents are talking about, e.g. "empire building", "preservation", "logical" and "training" on p 7 line 14 onwards, and guideline "negativity" line 55.

In the section about Continuity p 8 line 14 onwards there are several linguistic ambiguities as to who says what, who felt what? And what is meant by negative views (line 32)?

The section about Interface is interesting, but consider whether it is related to the research question. The slowness of hospital letters is well known - does it add something to the paper? p 9 line 9-17 seems out of analytical focus and should be removed. The same applies to the whole section about Practice-based follow-up. Screening is not mentioned in the research question.

The section about Prevention versus target driven care applies to a UK-context only - consider explaining more or removing it. The citation about public health disconnect (p 10 line 3) - how relevant is it to the research question?

Further on P 10 line 13 - 31 - same concern as above.
11.

The discussion could be developed further. The core theme is not discussed at all - what does 'managing risk' in primary care mean - and how does the concept relate to international literature? Which key issues for ca diagnosis in pc are highlighted by the findings of this study, and which theoretical perspectives could be developed from them?

Why is it difficult to diagnose cancer in primary care? If discontinuity in primary care could be a cause for diagnostic delay - should it be addressed by the campaigns? On p 13 lines 11 - 19 several suggestions are made, without discussion. Lines 26 - 41 have limited relevance to the research questions and they repeat points from the results without adding much.

12.

The strength of the core theme is not weakened by the sample size, but it should be validated by comparison to the international literature.

Thanks for the opportunity to read and comment on this interesting paper. Generally, I think that the language of the paper should express more curiosity and a critical attitude, and be less assertive. For example, the first sentence of the introduction contains four assertions and a causative linkage between them. If there is evidence for all this, fine, otherwise the credibility of the paper is weakened already from the start. Similarly, the last sentence of the paper seems to be speculative. If you think that the results warrant such a conclusion, the claim should be supported by a line of arguments.

The authors claim to have opened up a new field of research. However, this new field draws on a range of already established research fields, which the authors need to be familiar with in order to lead an adequate discussion: Diagnostic reasoning and medical decisionmaking, knowledge management in primary care, the concept and definitions of 'delay', patients' experience of help seeking and so

	on. The introduction addresses the national UK-situation only. As BMJ Open is an international journal, the perspective and reference should be wider. The Methods section designate the sampling as random, however, it is strategic. Please consider moving the list of participants to the Methods section. The percentages are not necessary. In general, the mixed staff perspective of the study could be more exploited. And finally: On the first title page, the third author is called Maria Dey instead of Paola.
--	---

REVIEWER	Thomas Round King's College London
REVIEW RETURNED	08-Mar-2014

GENERAL COMMENTS	Overall an interesting paper, but I think some changes need to be made before publication. I will outline some of the suggested changes below. Introduction There is a good overview of some of the recent quality improvement initiatives for cancer. There is not enough outline of why this study is unique in the introduction/overview of the literature. There needs to be more of an overview of any previous qualitative research in this area and why this study is unique. There have been some qualitative studies in this area, which would be worth referencing and then stating why this study is unique. Examples include: Mitchell 2012 qualitative paper in Australia, is about GPs role in cancer follow up, but not much on early diagnosis/referral. http://onlinelibrary.wiley.com/doi/10.1111/j.1365-2524.2012.01075.x/full Farquhar 2005 Paper on Ovarian cancer- small amount on diagnostic stage, but more on interface with secondary care http://onlinelibrary.wiley.com/doi/10.1111/j.1365-2354.2005.00596.x/pdf Daly 2007, a mixed qual/quant study on Barriers to early diagnosis in primary care (Ireland). Focus groups. From the qualitative research: The principle barriers identified were delayed patient presentation, lack of direct GP access to radiological and endoscopic investigations, difficulty with referral of patients to hospital services, lack of clear recommendations for cancer screening, poor communication with hospital services and inequitable access to hospital services for patients who cannot afford to pay privately. These were not confined to early diagnosis but applied to cancer diagnosis at any stage. http://www.ncbi.nlm.nih.gov/pubmed/18277732 http://www.imj.ie/ViewArticleDetails.aspx?ArticleID=2677 Methods More information needs to be provided in the methods section. Why was the decision made to have both individual and group interviews? Was a topic guide used? le how was the interview semi-structured. It is discussed how GP practices in the area were stratified into 3 groups (high to low engagers) and then a random sample of 10 were approached. It would be useful to know how
--

many practices were in the LSSCN at the time, and how many were assessed into each engagement category. I also wonder if at some point the authors should comment on this high to low engagement stratification, was this a somewhat arbitrary stratification? What were the other characteristics of practices in the different stratifications, as quality improvement related to cancer in primary care should not be viewed in isolation.

There needs to be more comment on the merging of the medium and low engagement groups. This could be in the results section (see later, as more information is needed about the practices who took part). It is also not clear how the group based interviews versus the individual interviews worked. In the practices with group interviews, was there just one interview? How many members attended? Was there a topic guide? Were these more of a focus group design? How long did the interviews go on for? Also again for the individual interviews how many and with whom occurred at each practice?

This could be as an appendix, outlining staff members and how many participated at each practice (or even a table in the results section with more information about practices and individual participants).

Results

As noted in the methods section there needs to be more information about the 6 practices who agreed to take part. This could be as a table or an appendix. This could include practice list size, number of full time equivalent GPs, socio-economics of practice performance, even eg practice QoF score etc. This is to give an overview of how representative these practices might be for the readers. There also needs to be a breakdown of the group versus individual interviews, and how many/who participated in each at each practice. Were the practices who were “low/medium” engagers all from one stratification eg were they all medium engagers? I think as noted should be some comment in this rather arbitrary classification. Was it done to see if there was any difference between these practices? Though this is not mentioned in the results/discussion.

Under “complexity” the authors write “whilst some symptoms were straightforwardly associated with cancer”. I think this should be re-phrased. Some clinical features should as “red flags” have higher predictive value for cancer, compared to other less specific features (ie low risk but no risk features). I am not sure the text starting “other symptoms commonly presenting in practice, compared with the relative infrequency of cancer” makes sense clinically. Cancer is a low incidence but serious disease for GPs. I am not sure the quote from interview 4 participant 1 about “what you’re supposed to be doing and referral pathways” makes sense in this context as GPs will utilise 2WW referrals not uncommonly, but only around 11% of all their suspected cancer 2WW referrals (PPV) will have cancer overall.

In regards to “fragmentation” the authors mention “rigid referral criteria led to cases being bounced back”, and participants “fudge” or “bend the rules”. Is this really fragmentation? Or more to do with the actual referral tools/guidelines themselves? Such as what level of predictive value cut off should the guidelines have. The authors also mention access to diagnostic services and restrictions. They comment on whether this is a logical delivery model, but did any participants mention cost aspects to expansion of access to diagnostics?

The sentence beginning “guidance that expressed the statistical probabilities that someone had cancer was less preferred” should be

	re-phrased. Continuity of care is mentioned, and that its absence “could contribute to diagnostic delay” although the literature on this is inconclusive. They mention that smaller, rural practices had potentially closer relationships with patients that could lead to earlier consultation, however as noted they do not outline the type of practices who participated in this study and whether any of these were smaller/rural practices. Discussion There are other qualitative studies as outlined previously around aspects of early cancer diagnosis, so they should offer an overview of how this study relates and is different to previous literature. There is also a qualitative study into GP use of risk assessment tools (CAPER) that would be worth referencing. The authors mention that practice staff highlighted that patient-doctor relationships were being eroded in large practices- but give no indication as noted whether the practices who took part in this study were small/large etc. They state that “receptionists could act as navigators directing patients with symptoms to see the GP”, but do not comment on the ethical/privacy implications of this. For example how easy is it for a patient to mention eg rectal bleeding to a receptionist, when potentially other patients are around? Therefore it is not just professional boundaries that could impede this but the ethical implications for patients and staff. The authors mention support for the 2WW referral initiative, and mention concerns regarding potential system delays ie for information to be sent back to the practice. But they do not mention what about patients who do not fit the 2WW criteria, who may have low risk but no risk symptoms – how does primary care manage this risk? What about any possible changes to the 2WW criteria? Is it possible to make conclusions about cancer related initiatives without reviewing the referral system criteria themselves? It would be interesting to know what was in the topic guide. As noted there needs to be more information on the practices who participated for external validity. It would also be useful to have some more specific/concrete conclusions.
--	--

VERSION 1 – AUTHOR RESPONSE

Overall an interesting paper, but I think some changes need to be made before publication. I will outline some of the suggested changes below.

Introduction

There is a good overview of some of the recent quality improvement initiatives for cancer. There is not enough outline of why this study is unique in the introduction/overview of the literature. There needs to be more of an overview of any previous qualitative research in this area and why this study is unique. There have been some qualitative studies in this area, which would be worth referencing and then stating why this study is unique.

Examples include:

Mitchell 2012 qualitative paper in Australia, is about GPs role in cancer follow up, but not much on early diagnosis/referral.

<http://onlinelibrary.wiley.com/doi/10.1111/j.1365-2524.2012.01075.x/full>

Farquhar 2005 Paper on Ovarian cancer- small amount on diagnostic stage, but more on interface with secondary care

<http://onlinelibrary.wiley.com/doi/10.1111/j.1365-2354.2005.00596.x/pdf>

Daly 2007, a mixed qual/quant study on Barriers to early diagnosis in primary care (Ireland). Focus

groups. From the qualitative research: The principle barriers identified were delayed patient presentation, lack of direct GP access to radiological and endoscopic investigations, difficulty with referral of patients to hospital services, lack of clear recommendations for cancer screening, poor communication with hospital services and inequitable access to hospital services for patients who cannot afford to pay privately. These were not confined to early diagnosis but applied to cancer diagnosis at any stage.

<http://www.ncbi.nlm.nih.gov/pubmed/18277732>

<http://www.imj.ie/ViewArticleDetails.aspx?ArticleID=2677>

Methods

More information needs to be provided in the methods section. Why was the decision made to have both individual and group interviews? Was a topic guide used? If so, how was the interview semi-structured. It is discussed how GP practices in the area were stratified into 3 groups (high to low engagers) and then a random sample of 10 were approached. It would be useful to know how many practices were in the LSSCN at the time, and how many were assessed into each engagement category. I also wonder if at some point the authors should comment on this high to low engagement stratification, was this a somewhat arbitrary stratification? What were the other characteristics of practices in the different stratifications, as quality improvement related to cancer in primary care should not be viewed in isolation.

There needs to be more comment on the merging of the medium and low engagement groups. This could be in the results section (see later, as more information is needed about the practices who took part). It is also not clear how the group based interviews versus the individual interviews worked. In the practices with group interviews, was there just one interview? How many members attended? Was there a topic guide? Were these more of a focus group design? How long did the interviews go on for? Also again for the individual interviews how many and with whom occurred at each practice? This could be as an appendix, outlining staff members and how many participated at each practice (or even a table in the results section with more information about practices and individual participants).

Results

As noted in the methods section there needs to be more information about the 6 practices who agreed to take part. This could be as a table or an appendix. This could include practice list size, number of full time equivalent GPs, socio-economics of practice performance, even eg practice QoF score etc. This is to give an overview of how representative these practices might be for the readers. There also needs to be a breakdown of the group versus individual interviews, and how many/who participated in each at each practice. Were the practices who were “low/medium” engagers all from one stratification eg were they all medium engagers? I think as noted should be some comment in this rather arbitrary classification. Was it done to see if there was any difference between these practices? Though this is not mentioned in the results/discussion.

Under “complexity” the authors write “whilst some symptoms were straightforwardly associated with cancer”. I think this should be re-phrased. Some clinical features should as “red flags” have higher predictive value for cancer, compared to other less specific features (ie low risk but no risk features). I am not sure the text starting “other symptoms commonly presenting in practice, compared with the relative infrequency of cancer” makes sense clinically. Cancer is a low incidence but serious disease for GPs. I am not sure the quote from interview 4 participant 1 about “what you’re supposed to be doing and referral pathways” makes sense in this context as GPs will utilise 2WW referrals not uncommonly, but only around 11% of all their suspected cancer 2WW referrals (PPV) will have cancer overall.

In regards to “fragmentation” the authors mention “rigid referral criteria led to cases being bounced back”, and participants “fudge” or “bend the rules”. Is this really fragmentation? Or more to do with the actual referral tools/guidelines themselves? Such as what level of predictive value cut off should the guidelines have. The authors also mention access to diagnostic services and restrictions. They comment on whether this is a logical delivery model, but did any participants mention cost aspects to

expansion of access to diagnostics?

The sentence beginning “guidance that expressed the statistical probabilities that someone had cancer was less preferred” should be re-phrased.

Continuity of care is mentioned, and that its absence “could contribute to diagnostic delay” although the literature on this is inconclusive. They mention that smaller, rural practices had potentially closer relationships with patients that could lead to earlier consultation, however as noted they do not outline the type of practices who participated in this study and whether any of these were smaller/rural practices.

Discussion

There are other qualitative studies as outlined previously around aspects of early cancer diagnosis, so they should offer an overview of how this study relates and is different to previous literature. There is also a qualitative study into GP use of risk assessment tools (CAPER) that would be worth referencing. The authors mention that practice staff highlighted that patient-doctor relationships were being eroded in large practices- but give no indication as noted whether the practices who took part in this study were small/large etc.

They state that “receptionists could act as navigators directing patients with symptoms to see the GP”, but do not comment on the ethical/privacy implications of this. For example how easy is it for a patient to mention eg rectal bleeding to a receptionist, when potentially other patients are around? Therefore it is not just professional boundaries that could impede this but the ethical implications for patients and staff.

The authors mention support for the 2WW referral initiative, and mention concerns regarding potential system delays ie for information to be sent back to the practice. But they do not mention what about patients who do not fit the 2WW criteria, who may have low risk but no risk symptoms – how does primary care manage this risk? What about any possible changes to the 2WW criteria? Is it possible to make conclusions about cancer related initiatives without reviewing the referral system criteria themselves? It would be interesting to know what was in the topic guide.

As noted there needs to be more information on the practices who participated for external validity. It would also be useful to have some more specific/concrete conclusions.

VERSION 2 – REVIEW

REVIEWER	May-Lill Johansen Department of Community Medicine UiT The Arctic University of Norway
REVIEW RETURNED	04-May-2014

GENERAL COMMENTS	Thank you for the opportunity to review this paper again. Congratulations to the authors for rewriting it into a much better report. I only have some minor comments. In the results section, p 13 line 55 under the heading Prevention versus target driven care: The sentence starting with “It is recognized that...” seems to be background information, and not a result. Consider moving it. In the first sentence of the discussion, consider using “symptoms associated with cancer” instead of “suggestive of cancer”. In the second sentence of the discussion, I miss a discussion of the sorting that GPs have to do; by NOT referring most of the people with “symptoms associated with cancer” – as otherwise the system would be clogged. This is a crucial part of managing risk. Generally the discussion should be linguistically re-edited, as there are several awkward sentences, unclear wordings and ambiguities. Be sure that the points referred from other papers are clear and correct. Good luck with the publication!
---

REVIEWER	Thomas Round King's College London Member of National Cancer Research Institute (NCRI) Primary Care Study Group, and RCGP representative at National Collaborating Centre for Cancer Management Board.
REVIEW RETURNED	01-May-2014

GENERAL COMMENTS	Many thanks to the authors for their work on this and addressing the majority of the reviewers comments. The paper is much improved in all sections, particularly the introduction and the methods. Thanks to the authors for improving the methods section, but I still feel needs some minor clarifications as some aspects are not clear. If these minor alterations are made then I would recommend publication (and these changes should not take much more time/effort). Please see these minor recommended edits below: The authors write "Interviews (group and individual) were held on only one occasion in each practice". They then write that at the 6 practices "Thirty nine participants took part in group (n=5) or individual (n=4) interviews". The statement that interviews were held on only one occasion doesn't seem to make sense with the 9 different interviews (mix of individual and group), and the text/explanation needs re-editing. This could be very simply be clarified by editing and expanding table 1 (and I also feel that the information on table 1 is not particularly helpful at present, eg should include total number of GPs (salaried and partners), and some idea of list size). This could be a practice based table, and describe each practice in more detail. For example: Practice 1: 4 GPs, practice list size (could be in categories eg <3000, 3000-5000, 5000-8000, >8000), Deprivation, population density. Number of individual and/or group interviews (with a breakdown of staff at the group interviews eg 1 GP, 2 nurses, 1 receptionist, 1 manager). This would then give the reader a clear representation of each practice, and who took part in the study from each practice. This aids with understanding and the external validity if the study. As noted if these minor changes are made to the methods and table 1 then I would recommend publication.
--

VERSION 2 – AUTHOR RESPONSE

Reviewer Name Thomas Round

The authors write "Interviews (group and individual) were held on only one occasion in each practice". They then write that at the 6 practices "Thirty nine participants took part in group (n=5) or individual (n=4) interviews". The statement that interviews were held on only one occasion doesn't seem to make sense with the 9 different interviews (mix of individual and group), and the text/explanation needs re-editing.

We have re-edited this to clarify the methods.

This could be very simply be clarified by editing and expanding table 1 (and I also feel that the information on table 1 is not particularly helpful at present, eg should include total number of GPs (salaried and partners), and some idea of list size). This could be a practice based table, and describe each practice in more detail. For example:

Practice 1: 4 GPs, practice list size (could be in categories eg <3000, 3000-5000, 5000-8000, >8000), Deprivation, population density. Number of individual and/or group interviews (with a breakdown of staff at the group interviews eg 1 GP, 2 nurses, 1 receptionist, 1 manager). This would then give the reader a clear representation of each practice, and who took part in the study from each practice. This aids with understanding and the external validity if the study.

While we sympathise with the reviewer, we have serious concerns that such a table would compromise the confidentiality agreed with the practices and participants particularly as the area of the country will be known from other information provided. We have therefore added a sentence in the table to that effect. We have also provided an additional sentence in the results to demonstrate the range of participants in each practice.

Reviewer Name May-Lill Johansen

In the results section, p 13 line 55 under the heading Prevention versus target driven care: The sentence starting with "It is recognized that..." seems to be background information, and not a result. Consider moving it.

This has now been removed.

In the first sentence of the discussion, consider using "symptoms associated with cancer" instead of "suggestive of cancer".

This has been amended.

In the second sentence of the discussion, I miss a discussion of the sorting that GPs have to do; by NOT referring most of the people with "symptoms associated with cancer" – as otherwise the system would be clogged. This is a crucial part of managing risk.

We have added a paragraph on this topic.

'The International Cancer Benchmarking Partnership has observed lower survival rates in Denmark and the UK compared to other countries in their study, which has raised concerns about the role of primary care gatekeeping in delayed diagnosis[2]. This observation appears to be supported by a wider study of European countries which also demonstrated lower cancer survival in those with primary care based gatekeeper systems.[18] Primary care gatekeeping may reduce the burden on specialist care but may also contribute to diagnostic delay through the restriction of access to these services. In the UK, national referral guidelines and the two-week waiting-time initiative should help alleviate this problem for those with symptoms associated with cancer.'

Generally the discussion should be linguistically re-edited, as there are several awkward sentences, unclear wordings and ambiguities.

We have re-edited this section as requested.

Be sure that the points referred from other papers are clear and correct.

We have gone through the references again and made minor changes to clarify the points raised by the papers and replaced one reference with a primary source.